# Fault Voiceprint Signal Diagnosis Method of Power Transformer Based on Mixup Data Enhancement

**DOI:** 10.3390/s23063341

**Published:** 2023-03-22

**Authors:** Shuting Wan, Fan Dong, Xiong Zhang, Wenbo Wu, Jialu Li

**Affiliations:** 1Hebei Key Laboratory of Electric Machinery Health Maintenance & Failure Prevention, North China Electric Power University, Baoding 071003, China; 2Hebei Engineering Research Center for Advanced Manufacturing & Intelligent Operation and Maintenance of Electric Power Machinery, North China Electric Power University, Baoding 071003, China; 3Department of Mechanical Engineering, North China Electric Power University, Baoding 071003, China

**Keywords:** power transformer, voiceprint signal, mixup, CNN

## Abstract

A voiceprint signal as a non-contact test medium has a broad application prospect in power-transformer operation condition monitoring. Due to the high imbalance in the number of fault samples, when training the classification model, the classifier is prone to bias to the fault category with a large number of samples, resulting in poor prediction performance of other fault samples, and affecting the generalization performance of the classification system. To solve this problem, a method of power-transformer fault voiceprint signal diagnosis based on Mixup data enhancement and a convolution neural network (CNN) is proposed. First, the parallel Mel filter is used to reduce the dimension of the fault voiceprint signal to obtain the Mel time spectrum. Then, the Mixup data enhancement algorithm is used to reorganize the generated small number of samples, effectively expanding the number of samples. Finally, CNN is used to classify and identify the transformer fault types. The diagnosis accuracy of this method for a typical unbalanced fault of a power transformer can reach 99%, which is superior to other similar algorithms. The results show that this method can effectively improve the generalization ability of the model and has good classification performance.

## 1. Introduction

A power transformer is a key component in the power system, its vibration and noise affect the operation of the power system [1,2]. Affected by substation environment and human factors, different kinds of faults such as Partial Discharge (PD), short circuit impact, and so on, often occur during operation. It will not only affect the normal social electricity demand but also cause incalculable losses. It is necessary to monitor and diagnose the running state of the power transformer in real-time.

At present, the fault detection methods for power transformers have many ways of diagnosis. The authors of [3] used CNN to classify six kinds of discharge defects in power transformers and solved the diagnosis problem when using an ultra-high-frequency (UHF) relief valve sensor. In [4], the problem of dissolving gas samples in power transformers was studied. The proposed CNN model improves the accuracy of a few fault categories. The authors of [5] established the fault tree model of the power transformer and put forward the calculation method of comprehensive fault rate to solve the aging and fault problems of an oil-immersed power transformer during operation. Based on this model, the reliability and safety of power transformers were actually improved. According to the five typical fault types of a power transformer, the authors of [6] divided the early, middle, and late fault conditions of a power transformer into three stages. For each stage, the method of combining dissolved gas-in-oil analysis (DGA) and infrared diagnosis was used to conduct in-depth research on its fault conditions, and the power transformer’s Fault tree provides a new method for fault monitoring of the power transformer.

However, the above methods need contact measurement with the power transformer shell when collecting fault characteristics, and the operation is complex and difficult. The voiceprint signal acquisition process is non-contact, simple, and fast, and does not affect the normal operation of the transformer, making the research direction based on voiceprint signal gradually become a hot topic. The authors of [7] transformed the preprocessed voiceprint signal into the Mel spectrum and used a convolutional neural network to classify various fault features, which can effectively identify the abnormal state of the power transformer. The authors of [8] classified voiceprint signals of 10 kV dry-type transformers under different faults based on the gammatone-frequency cepstral coefficients (GFCC) voiceprint spectrum. The authors of [9] used a UHF PD sensor to capture four typical PD mode UHF signals. After preprocessing, a convolutional neural network was used to classify samples, which improved the accuracy of PD pattern recognition. However, the above deep learning algorithms need to provide a large number of learning samples to improve the training accuracy of the model. However, low-probability fault events such as short circuit impact the lack of enough samples for experimental research. When the fault samples are unbalanced, it is easy to have over-fitting events, which affects the classification effect of the model.

For the problem of unbalanced sample size, data enhancement is often used to expand the sample. The authors of [10] adopted the least-square GAN (LS-GAN) method to expand the data in different proportions, which significantly improves the performance of the classifier. The authors of [11] used SinGAN to generate additional scale maps, which provides a good data basis for the subsequent long short-term memory (LSTM) model. The authors of [12] proposed a data enhancement method for short-term voltage stability assessment (STVSA), which has achieved good results in processing small sample datasets. The authors of [13] used an over-sampled technology to balance the DGA data of the power transformer, and the effect was significantly improved compared with the original dataset. The above methods have achieved good expansion effects for unbalanced data. Unfortunately, there is no corresponding data enhancement method based on the imbalance of power-transformer voiceprint signal samples.

Aiming at the imbalance of power-transformer fault categories, this paper proposes a method of power-transformer fault voiceprint signal diagnosis based on Mixup data enhancement. First, the Mel filter is used to reduce the dimension of various fault voiceprint signals to obtain the corresponding Mel spectrogram. Then, the Mixup algorithm is used to expand the data of the fault set with insufficient samples. Finally, CNN is used to classify the extended results. The problem of class imbalance in the training process is solved, the occurrence of over-fitting events is avoided, and the classification and diagnosis of typical fault voiceprint signals of power transformers are realized.

## 2. Voiceprint Signal Preprocessing and Classifier

### 2.1. Voiceprint Time Spectrum

The voiceprint time-frequency spectrum is the relationship between frequency and time obtained by processing the collected time-domain signal, which represents the energy distribution of the sound signal in the process of transmission [14,15]. The color depth of the spectrum shows the size of the signal energy. Generally, the darker the color, the stronger the energy contained in the signal.

The rendering process of the voiceprint time-frequency spectrum includes operations such as framing, windowing, and discrete Fourier transform. Figure 1 shows the complete rendering process of the voiceprint time spectrum. In this paper, the voiceprint signals of the power transformer in various states are cut into samples with a duration of 2 s, and the overlapping segmentation method is used for frame division. The frame length is uniformly set to 0.25 s, and the overlap rate is set to 50%. The window function is a Hamming window, as shown in Equation (1) [16]:(1)$$w(n)=\left\{\begin{array}{l} 0.54+0.46\cos(\frac{2\pi n}{N-1}),0\leq n\leq N-1\\ 0\;\;\;\;else \end{array}\right.$$
where *N* represents the length of the Hamming window.

### 2.2. Mel Time Spectrum

At present, many methods have emerged in the field of time-frequency analysis. Wavelet and Empirical Mode Decomposition (EMD) are commonly used time-frequency analysis methods. They obtain the corresponding time-frequency signal through the improvement of the Fourier transform, but the generated time-frequency spectrum data size is too large, which easily affects the classification efficiency of the subsequent depth learning algorithm. The Mel time spectrum can greatly reduce the sample size and improve the training effect of the subsequent classification model on the premise of preserving the voiceprint features. Therefore, this paper uses the Mel filter to reduce the dimension of the time spectrum of the power transformer fault voiceprint signal. The conversion relationship between the two scales is shown in Equation (2):(2)Mel(k)=2595×lg(1+f/700)
where *f* is the frequency at the conventional scale, 0 ≤ *f* ≤ 8000; *k* is the frequency under the Mel scale, Hz.

Human beings can easily distinguish voiceprint signals in the low-frequency range but are often insensitive to sound in the high-frequency range. To find the voiceprint features suitable for human resolution, the common frequency scale is often converted into the Mel frequency scale to form a linear relationship between the human ear’s perception of frequency and make it easier to detect the frequency changes of high-frequency voiceprint signals [17,18]. The typical fault voiceprint frequency of power transformer mainly occurs in medium and low frequencies below 2000 Hz. According to the fault characteristics, the frequency weight in the filter bank is adjusted, so as to realize the feature extraction of the fault voiceprint signal [19]. The Mel filter bank used in this paper is shown in Figure 2. The transfer function of the filter bank is shown in Equation (3):(3)$$H_{m}(f)=\left\{\begin{array}{l} 0,f<x(m-1)\\ \frac{f-x(m-1)}{x(m)-x(m-1)},x(m-1)\leq f\leq x(m)\\ \frac{x(m+1)-f}{x(m+1)-x(m)},x(m)\leq f\leq x(m+1)\\ 0,f>x(m+1) \end{array}\right.$$
where *m* is the filter bank number, the number of filters in this paper is set to 30, and *x* (*m*) is the center frequency of the triangular filter, as shown in Equation (4):(4)$$x(m)=(\frac{C}{f_{s}}){\mathrm{Mel}}^{-1}(\mathrm{Mel}(f_{\min})+m\frac{\mathrm{Mel}(f_{\max})-\mathrm{Mel}(f_{\min})}{M+1})$$
where *f*_max_ and *f*_min_ are the maximum and minimum frequency values of the filtering range; *f_s_* is the sampling frequency of the acoustic sensor, which is taken as 16 kHz; *C* is the frame length of the discrete Fourier transform.

We multiply the time-frequency spectrum matrix and the Mel filter bank matrix and take the logarithm to obtain the Mel time-frequency spectrum of the voiceprint signal in various states of the power transformer, as shown in Figure 3. Mel time-frequency spectrum characteristics of different faults are compared, and data compression and dimensionality reduction effect are good, which provides a good training basis for the following training process of CNN.

### 2.3. Mixup Data Enhancement

The purpose of data enhancement is to alleviate the imbalance and missing data in depth learning. After it was proposed, it was first widely used in the field of image processing and achieved good results. Its main direction is to increase the diversity of data in the process of deep learning training, so as to improve the generalization ability of the model. Today’s data enhancement methods can be divided into three categories: shape scaling and flipping, noise, and sampling. Scaling and flipping the image is the most original data enhancement algorithm. By changing the simple shape, the target data can be expanded infinitely. The method of adding noise to the image can further expand the image without changing the basic shape of the image. The image expanded by adding noise can also enhance the robustness of the model in the training process. By mastering the distribution of original data, sampling new data as a method of enhancing data has been widely used in the recent years. For the position and chromaticity deviation in the training data, geometric transformation and other methods are very good solutions. Its simple operation and easy implementation make it have unique advantages in enhancing many image libraries. However, in many fields such as power-transformer fault image analysis, the deviation between the same fault data is more complex than the difference between simple position and color change. Therefore, the scope of basic data enhancement is relatively limited.

Mixup [20] is a data enhancement method applied in the field of image classification, which is simple and effective. On the basis of basic data enhancement, this method synthesizes new images by randomly sampling samples from the training set and linear blending to achieve the purpose of data expansion [21]. It has good effects in many fields such as image, text, voice, etc.

The Mixup data enhancement method achieves the effect of data expansion by randomly extracting two sets of images and corresponding tags from the training set and fusing them to generate false samples. For an image, this means merging two different image pixels. It can be used as a regularization method during model training. The calculation process is shown in Equation (5) [22]. The two Mel time-frequency maps in a normal state are fused and expanded into a new spectrum. The new sample generated by the fusion contains the characteristics of two sets of original samples, but it is not the same as the original sample. In essence, it means to linearly interpolate two random samples, enhance the linear expression between samples, and improve the generalization performance of the model [23]. Figure 4 shows the processing results of Mixup on normal samples and three fault samples, in order to facilitate CNN’s classification training of the spectrum; all the coordinate parameters of the spectrum are removed, and only the image part of the Mel time spectrum is reserved, and the fusion coefficient is set to 0.5.
(5)$$\left\{\begin{array}{l} \tilde{x}=\lambda x_{i}+(1-\lambda )x_{j}\\ \tilde{y}=\lambda y_{i}+(1-\lambda )y_{j} \end{array}\right.\;\;\lambda \in [0,1]$$
where (*x_i_*,*y_i_)*(*x_j_*,*y_j_*), respectively, represent the eigenvectors and marker values of any two samples *i* and *j* in the sample set; ($\tilde{x}$,$\tilde{y}$) represents the eigenvector and marker value of the expanded sample; λ represents the fusion coefficient.

### 2.4. CNN

CNN is a multi-layer data perceptron designed for recognizing two-dimensional spatial features. Compared with other deep learning structures, a convolutional neural network can give better results when dealing with image problems. The features of samples at various levels are extracted mainly through convolution and pooling operations [24].

The main function of the convolution layer is to perform convolution calculations on the convolution core and the imported dataset to extract the corresponding features. By extracting special information from the input image, this information is called image features, such as the texture and color features of the image. These features are reflected by each pixel in the image through a combination or an independent way. The convolution operation is shown in Equation (6):(6)$$X_{j}^{e}=f(\sum_{i\in M_{j}}x_{i}^{e-1}\times k_{ij}^{e}+b_{j}^{e})$$
where, $X_{j}^{e}$ and $x_{i}^{e-1}$, respectively, represent the output and input characteristics of the layer *e* network, $M_{j}$ represents the set of input characteristics, f(·) represents the activation function, $k_{ij}^{e}$ represents the weight matrix of the convolution kernel, and $b_{j}^{e}$ represents the offset term in the convolution operation.

The convolution layer plays the role of feature extraction in the CNN model, but only the convolution processing of image data cannot reduce the number of image features. In the final fully connected layer, the model still faces the problem of a large number of parameters; so, the pooling layer is required to reduce the number of image features. The purpose of the pooling layer is to reduce the convolution kernel to achieve the purpose of dimensionality reduction [24,25,26,27]. At present, there are maximum pooling and average pooling. The pooling operation is performed by an *n* × *n* matrix window of *n* size slides, to calculate and reduce the size of the model by calculating the maximum and average values in the matrix window. The purpose of the maximum pooling layer is to take the maximum value in the specified image window as the output, while the average pooling layer is to take the average value of all values in the specified image window as the output. The pooling layer can speed up training and prevent over-fitting. The maximum pooling layer is shown in Equation (7):(7)$$P_{ij}=\underset{k\in U_{ij}}{\max}\;a_{k}$$
where $P_{ij}$ represents the maximum output value in the pooling operation, $U_{ij}$ represents the pooling area, $a_{k}$ represents the output matrix of the convolution operation.

The pooled layer is followed by the fully connected layer. The fully connected layer converts all the feature matrices of the pooled layer into one-dimensional feature vectors, which are generally placed at the end of the CNN structure, and are used to classify the feature matrices after multi-layer convolution pooling [28]. The process data from the pooling layer to the fully connected layer will be mapped from more to less to reduce the dimension. To prevent the over-fitting phenomenon, the Dropout operation is carried out at the fully connected layer; it can reduce the complex coadaptation relationship between neurons and further enhance the generalization performance of the model [29]. 

The selection of structural parameters of the CNN model directly affects the classification ability of the model. In order to obtain the optimal results, the model structural parameters need to be optimized. In this paper, the optimization of CNN structure parameters is based on the grid search method [30]. This method lists different parameter combinations through the exhaustive method; through a circular traversal in the training set, the selected optimal parameters can maximize the accuracy of the model, especially when dealing with small sample data, and it has a significant effect. The CNN structure parameters constructed in this paper are shown in Table 1.

## 3. Voiceprint Signal Diagnosis Process

The collected voiceprint signal of the power transformer is a relatively complex mixed signal. When the equipment fails, its voiceprint characteristics are often affected by other equipment’s operation sound and transmission-path coupling interference factors. The fault types are complex and the probability is unbalanced. Simple fault diagnosis methods are not useful to accurately extract fault information from the collected signals and classify them one by one. The data processing method of Mel time-frequency spectrum combined with Mixup data enhancement proposed in this paper solves the characteristics of complex and unbalanced fault types. The pattern recognition of different fault types can be realized by combining the CNN deep learning algorithm. The diagnosis process is shown in Figure 5.

## 4. Experiments and Results

### 4.1. Source of Experimental Data

To verify the fault identification effect of CNN on the voiceprint signal of a power transformer, the acoustic signal acquisition equipment is used to collect the long-period acoustic signal of several 110 kV transformers in a substation. The acquisition time is 360 days. The layout of sensors and data storage units is shown in Figure 6, the frequency response range is 10 Hz~20 kHz, and the sampling frequency is set at 16 kHz. The acquisition environment is the outdoor substation environment under the normal state, and there is non-periodic noise interference such as wind noise and bird song. The audible voiceprint signals of the short circuit impulse, partial discharge, DC bias fault, and normal state are collected.

In the working process of a power transformer, the probability of short circuit impact and other faults is small, which easily causes the problem of insufficient experimental research samples and the unbalanced number of various samples. This paper studies the limited fault dataset of a power transformer. The corresponding voiceprint time spectrum and Mel time spectrum are drawn, respectively. In addition, basic data enhancement and mixup data enhancement are carried out for the acquired spectrum. The number of short circuit impact faults has increased from 10 to 600, the number of samples in normal state and two other kinds of faults is high; the number of samples expanded from 50 to 600, and 3/4 of the samples are selected for model training and 1/4 for test samples. In order to verify the accuracy of the expanded data, the original data are added to the test set for classification and identification. The results are shown in Table 2.

### 4.2. Experimental Result

The four constructed datasets are, respectively, imported into the CNN model for training. The CNN model is built under the Python environment using the deep learning framework Tensorflow. The computing platform is configured as Intel i7-8750H, CPU 2.20 GHz, 16 GB of memory, and Win10, 64-bit operating system. The number of iterations is 50, and the accuracy of fault sample classification is calculated. The results are shown in Figure 7. In Figure 7a, the algorithm model of using mixup to enhance Mel’s spectrum has the best training results. After 20 training epochs, the model tends to be stable and the classification accuracy is the highest. However, in Figure 7b, the accuracy of the model using the basic enhancement algorithm decreased during the test, and there was a slight over-fitting. It is proved that the simple method of flipping and color matching has certain defects in processing spectral images, and the mixup data enhancement method can enhance the generalization performance of the model. Figure 7c,d shows the results of data enhancement using a spectrogram. The accuracy of the two models fluctuates greatly, and the models using basic data enhancement have obvious over-fitting compared with the mixup algorithm. Compared with Figure 7a,b, it shows that the dataset constructed by the Mel time-frequency spectrum has strong robustness, which improves the training speed and diagnosis effect of the model.

The confusion matrix is the most intuitive and simple method to measure the accuracy of the classification model. It can clearly summarize and display the training effect of the test set and is an important indicator to evaluate the effect of the deep learning model. In order to show more clearly the difference between the accuracy of the four algorithms when using CNN classification, the confusion matrix is used to represent the accuracy of the four datasets. The calculation results are shown in Figure 8. The mel-mixup algorithm accurately classifies the voiceprint data of transformers in different states, the overall classification accuracy is up to 99%, and the diagnosis effect is good. The second is the Mel-Basic data enhancement algorithm, which has an accuracy rate of 95.5%. The accuracy rate of the dataset using the Time-frequency method is lower than that of the former, while the accuracy rate of the dataset using Time-frequency-Basic data enhancement is only 90%, which proves the limitations of the basic enhancement algorithm in processing spectral data.

To avoid the randomness of the results caused by a single test, the four datasets generated were repeated five times. The experimental results are shown in Figure 9. The diagnostic accuracy of the Mel-mixup algorithm has exceeded 98%, and the accuracy of the two experiments has reached 100%. The accuracy of basic data enhancement combined with the Mel time-frequency diagram method is about 96%; compared with the model using mixup as the data enhancement algorithm, it has decreased by about 3%. The accuracy of the model using the voiceprint time-frequency diagram as the dataset fluctuates greatly between several tests, and the accuracy is below 95%. The accuracy of the basic data enhancement method is around 90%. This proves that the dataset formed by the Mel-mixup algorithm when processing the voiceprint signal of a power transformer fault has good generalization performance.

In order to verify the accuracy of data enhancement, the mixup-enhanced Mel time-frequency spectrum dataset and the original dataset without data enhancement are imported into the CNN model for classification testing, and the classification effect of the model is observed with the loss curve as the measurement standard. As shown in Figure 10, the dataset model without data enhancement cannot converge all the time, and the loss curve of the test set fluctuates greatly, falling into serious over-fitting. The loss curve of the model after data enhancement tends to be stable quickly. It is proved that a small amount of training data is difficult to provide enough information for the model. The correctness of a small amount of data will have a great impact on the results, and it is difficult to measure the quality of the model.

In order to further prove the superiority of this algorithm, we introduce SVM, RNN, and KNN classifiers, and observe the classification effect of each algorithm with accuracy as a measure. The classification effect is shown in Figure 11. The CNN algorithm has the highest classification accuracy. Compared with RNN algorithm, the accuracy is 5.75% higher, among them, SVM and KNN algorithms are not effective in signal diagnosis. Compared with the CNN algorithm, the accuracy rate decreased by 28.5% and 33.5%, respectively. It shows that these two algorithms are sensitive when dealing with the frequency spectrum of power transformer Mel, and the classification effect is poor. It can be seen that the CNN algorithm has good diagnostic efficiency for the voiceprint signal of the power transformer.

The fusion coefficient of the Mixup algorithm is randomly selected between [0,1], to explore the impact of the fusion coefficient on the experimental results. We calculate the fusion coefficient of the mixup algorithm separately when λ=0,0.1,⋯1 or a random value. The calculation results are shown in Table 3. When the value is 0.5, the accuracy, precision, recall, and F1 values of the model are 99.7%, 99.8%, 99.7%, and 99.7%, respectively. All indicators are better than other models with random values. As the value of λ approaches both sides, the accuracy of the model is gradually declining. When the value of λ is 0 and 1, that is, the parameter indexes reach the lowest level, then there is no Mixup fusion. It is proved that when processing power transformer’s spectrum-type pictures, unifying the two groups of pictures can improve the diagnosis ability of the model. It is proved that when using mixup to enhance the Mel time spectrum, uniformly fusing the corresponding two sets of spectrograms on a 1:1 scale can improve the robustness and diagnosis ability of the model.

## 5. Conclusions

Aiming at the imbalance of power-transformer fault categories, this paper proposes a voiceprint fault diagnosis method based on mixup data enhancement, which realizes data enhancement and classification diagnosis of power-transformer fault samples, and draws the following conclusions:The operating environment of a power transformer is complex and prone to various faults. Since some faults are prone to low probability and few samples exist, it is necessary to enhance and expand the fault set with insufficient samples to make the diagnosis model have good generalization performance. Simple basic data enhancement methods such as flipping and color matching have certain defects when processing spectral images. The dataset enhanced by mixup can increase the robustness of the learning model.Compared with the voiceprint time-frequency map, in the Mel time-frequency map, it is easier to distinguish the frequency change of the frequency voiceprint signal. It can greatly reduce the data size of the sample and facilitate the feature extraction of the subsequent depth learning algorithm.Mel time-frequency diagram of a power transformer enhanced by mixup data has a good effect when using the CNN algorithm for classification diagnosis. It improved the generalization ability of the model and the diagnostic accuracy is up to 99%. Compared with other similar deep learning algorithms, it has good diagnostic performance for power-transformer fault voiceprint signals.

However, due to the lack of pertinence and real-time intervention in fault voiceprint signal acquisition, there are only a few types of power transformer faults involved in the article. Other kinds of faults occasionally occur in the actual production work of the substation, resulting in the limitations of the algorithm. In a follow-up study, we will record and study the voiceprint fault of other power transformers, further improve the voiceprint fault dataset of power transformer, and improve the diagnosis effect of the power transformer.

## Figures and Tables

**Figure 1 sensors-23-03341-f001:**
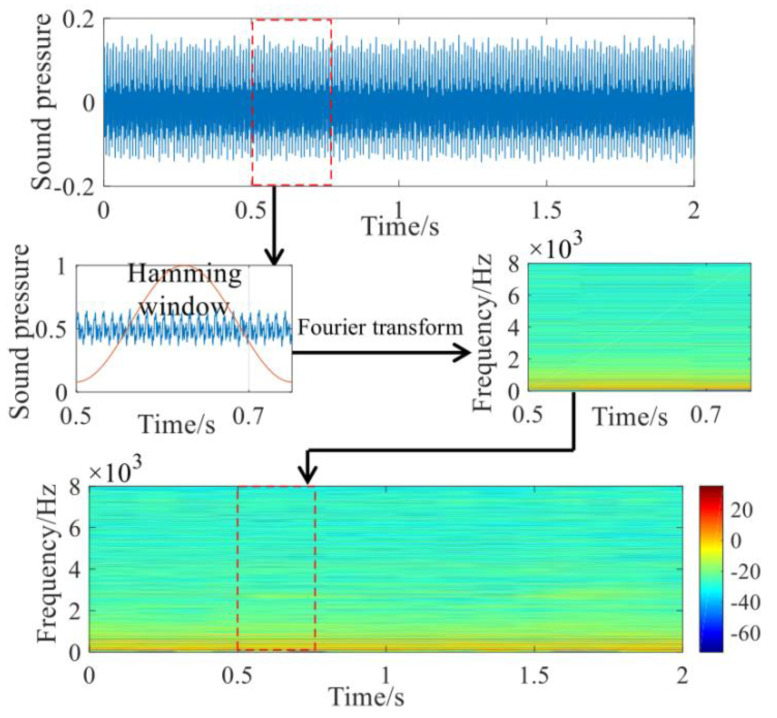
Voiceprint time spectrum drawing process.

**Figure 2 sensors-23-03341-f002:**
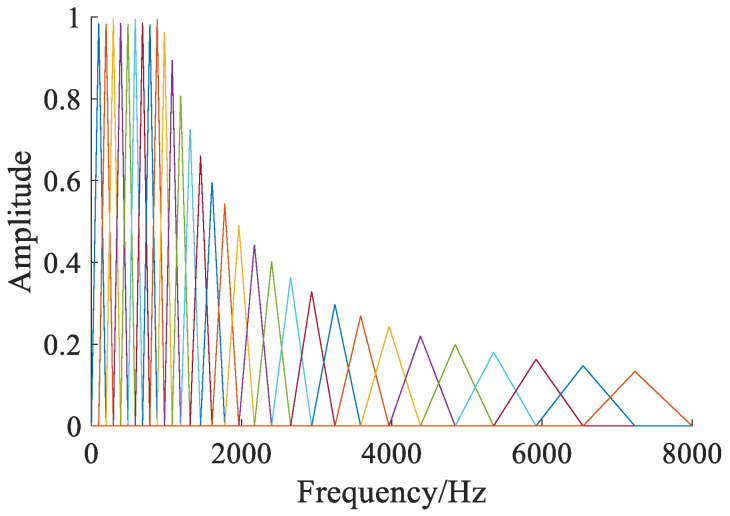
Mel filter bank.

**Figure 3 sensors-23-03341-f003:**
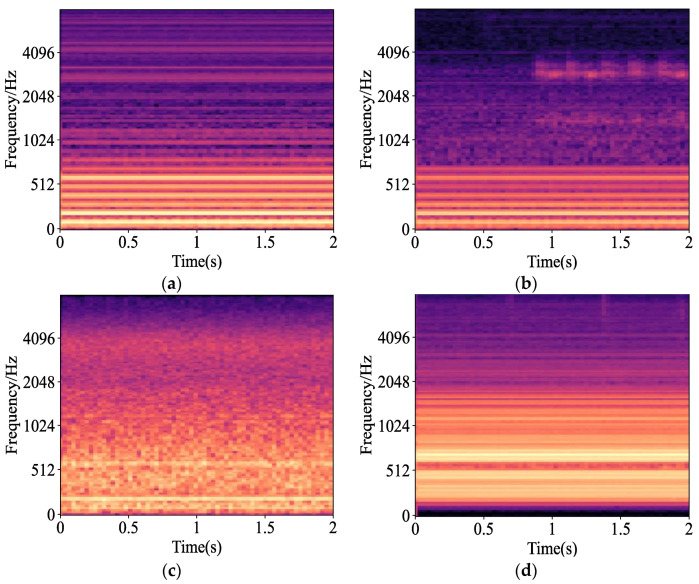
Mel time spectrum of a typical fault: (**a**) Normal; (**b**) Short-circuit impulse; (**c**) Partial discharge; (**d**) DC bias.

**Figure 4 sensors-23-03341-f004:**
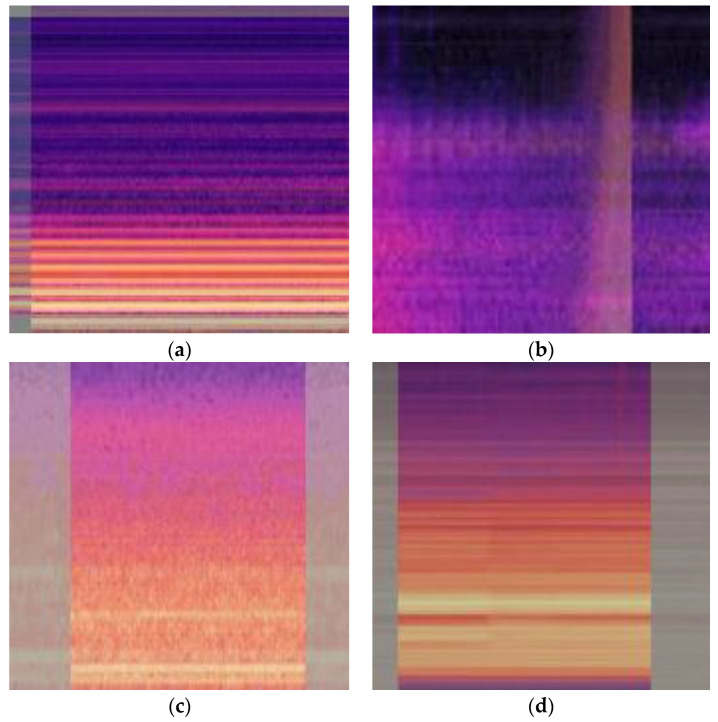
Mixup enhancement diagram: (**a**) Normal; (**b**) Short-circuit impulse; (**c**) Partial discharge; (**d**) DC bias.

**Figure 5 sensors-23-03341-f005:**
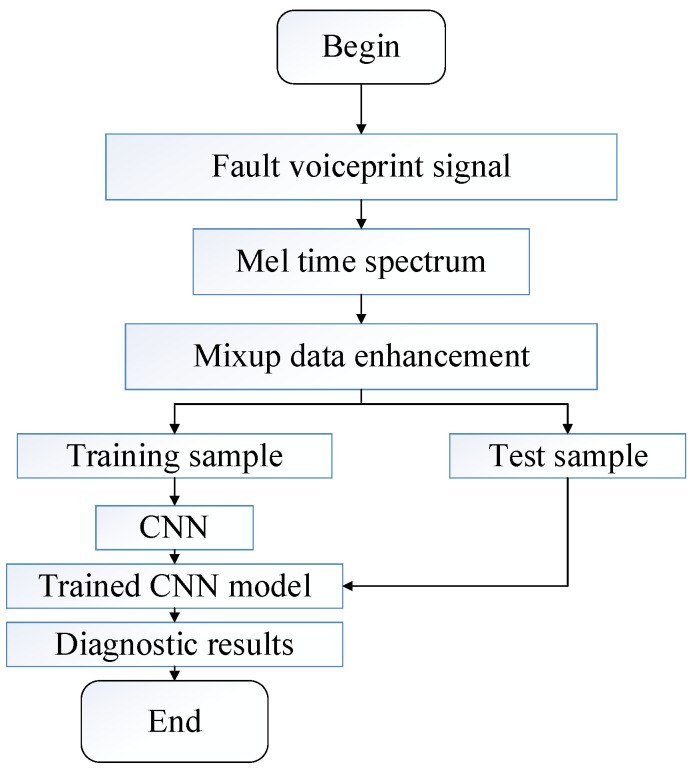
Diagnostic flowchart.

**Figure 6 sensors-23-03341-f006:**
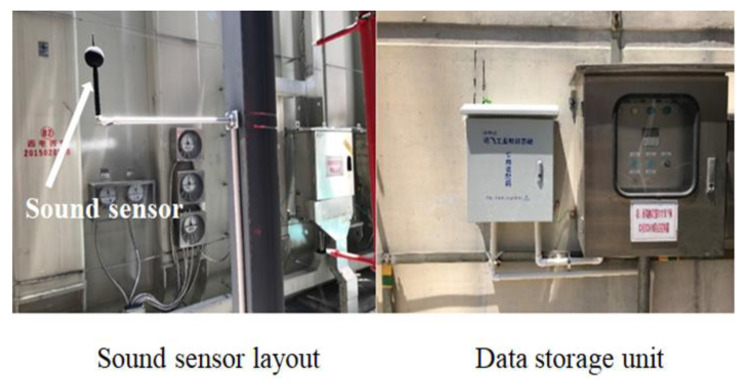
Field test environment.

**Figure 7 sensors-23-03341-f007:**
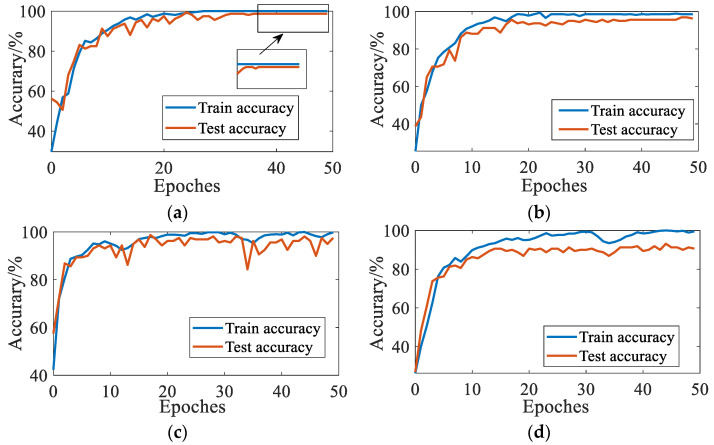
Diagnostic accuracy of different models. (**a**) Mel-Mixup; (**b**) Mel-Basic data enhancement; (**c**) Time-frequency-Mixup; (**d**) Time-frequency-Basic data enhancement.

**Figure 8 sensors-23-03341-f008:**
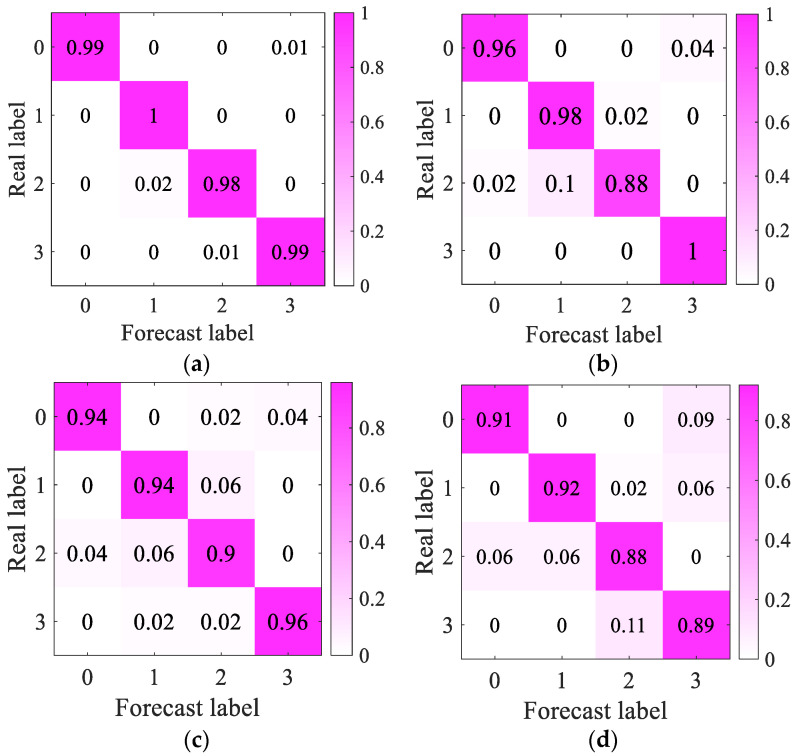
Fault classification confusion matrix. (**a**) Mel-Mixup; (**b**) Mel-Basic data enhancement; (**c**) Time-frequency-Mixup; (**d**) Time-frequency-Basic data enhancement.

**Figure 9 sensors-23-03341-f009:**
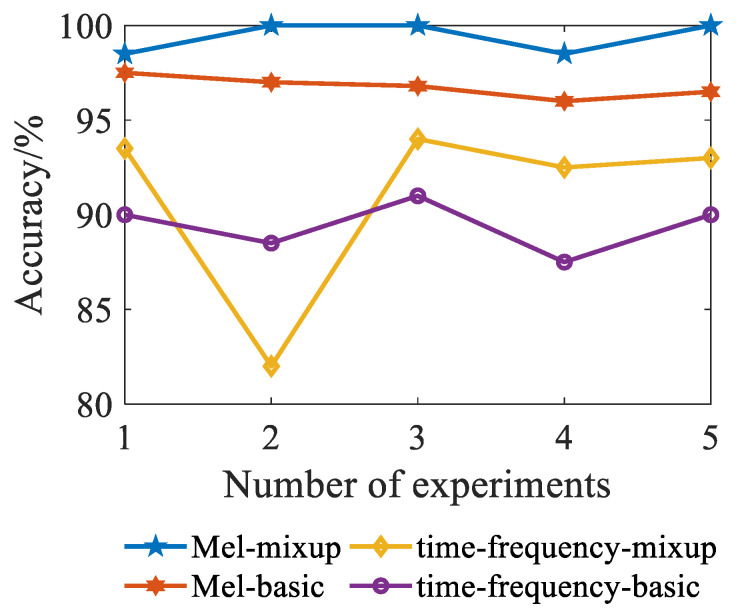
Comparison of diagnosis results of different models.

**Figure 10 sensors-23-03341-f010:**
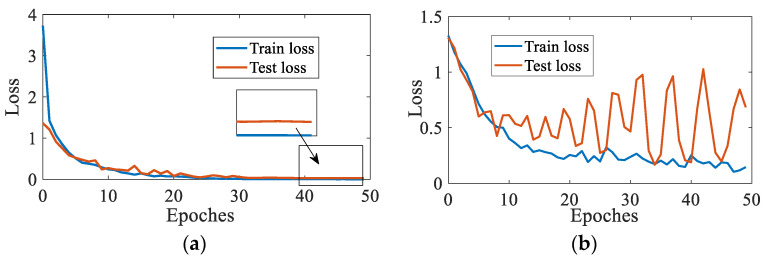
Loss curve (**a**) with data enhancement; (**b**) no data enhancement.

**Figure 11 sensors-23-03341-f011:**
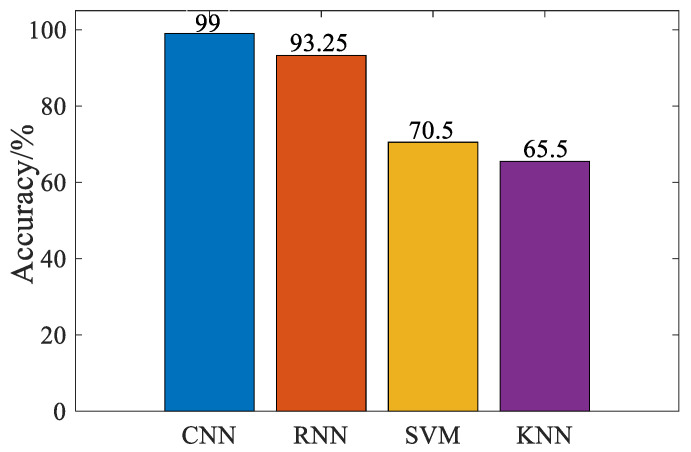
Classification accuracy performance of different methods.

**Table 1 sensors-23-03341-t001:** CNN Model Parameters.

Network Layer	Number and Size of Convolution Kernels	Step	Network Layer Output
Convolution layer 1	32@3 × 3	1	32@171 × 170
Pool layer 1	3 × 3	3	32@57 × 56
Convolution layer 2	64@3 × 3	1	64@55 × 54
Pool layer 2	3 × 3	3	64@18 × 18
Convolution layer 3	128@3 × 3	1	128@16 × 16
Pool layer 3	3 × 3	3	128@5 × 5
Full connection layer 1	128		128
Full connection layer 2	4		4

**Table 2 sensors-23-03341-t002:** Classification of experimental datasets.

Fault Number	Fault Type	Training Set	Test Set
0	normal	450	150
1	Short-circuit impulse	450	150
2	partial discharge	450	150
3	DC bias	450	150

**Table 3 sensors-23-03341-t003:** Effect of Mixup fusion coefficient on model performance.

λ	Accuracy/%	Precision/%	Recall/%	F1/%
0	96.5	96.8	96.5	96.6
0.1	97.7	97.8	97.7	97.7
0.2	98.1	99.0	98.1	98.5
0.3	98.1	98.3	98.1	98.2
0.4	98.3	99.1	98.3	98.7
0.5	99.7	99.8	99.7	99.7
0.6	99.1	99.2	99.1	99.1
0.7	98.1	99.0	98.1	98.5
0.8	97.5	97.8	97.5	97.7
0.9	97.1	97.2	97.1	97.1
1.0	96.5	97.0	96.5	96.7
random	98.1	98.3	98.1	98.2

## Data Availability

Not applicable.

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
