# Peer review of "Fault Voiceprint Signal Diagnosis Method of Power Transformer Based on Mixup Data Enhancement"

_sensors, 2023, doi:10.3390/s23063341_

Round 1

Reviewer 1 Report

This paper presents a method of voiceprint signal diagnosis for typical faults of power transformer based on mixup data enhancement. In general, the topic is timely and interesting, and the paper is well written. However, there are still some concerns of this reviewer:

1 How to validate the accuracy of the proposed data augmentation? 

2 Use of data augmentation for deep learning-based pattern recognition is a key idea in this paper. Authors should include and review some recently published papers with the similar scenario and research targets to enable readers to better know the latest developments in the field. Please refer to: A deep-learning intelligent system incorporating data augmentation for short-term voltage stability assessment of power systems; Incipient fault diagnosis in power transformers by data-driven models with over-sampled dataset.

3 The proposed method might be sensitive to the values of its main control parameters.  How did you tune the parameters? Please clarify. 

4 Please specify details of the computing platform and programming language used for this study. 

5 What are the limitations of this work in practical applications? Please elaborate on it. 

6 The full names of the abbreviations and terminology such as “DGA” and “UHF” should be given when they first appear.

7 Although the manuscript is well written in terms of English, there are some (very few, indeed) grammatical and expression errors. It is suggested to proofread the paper.

Reviewer 2 Report

Authors proposed methodlogy titled as " A Method of Voiceprint Signal Diagnosis for Typical Faults of Power Transformer Based on Mixup Data Enhancement".There are several changes and justifications required in manuscript.My comments are as follows:

1. Title should be reframed. A method..., typical faults.....mixup data... are unclear.

2. Abstract should include numerical results and should be restructured for better clarity.

3. Authors mentioned about imbalance data for classification.It can be address by using GAN.Suggested to discuss in revised version with following reference:

a. https://link.springer.com/article/10.1007/s00170-022-09356-0

b. https://www.mdpi.com/2227-9709/10/1/28

4. Fig.1. is unclear to readers.Here authors mentioned about Fourier Transform.Kindly clear the methodology.Authors applied Mel time algorithm or Fourier Transform.

5. Mel time frequency analysis is a combined time frequency analysis.There are other time frequency analysis methodlogy like Wavelets,EMD etc. Why author have chosen Mel time as compared to Wavelet and EMD.Kindly add discussion in revised version.To get more idea refer 3a.

6. In Table 1,authors mentioned CNN mdoel parameters.How the values are selected.Kindly address.

7. It is always a good practice to compare the classification results with different algorithms.It is strongly recommended to compare results with other DL models.

8. Conclusion section should be strengthened.

9. Future scope and limitations of methodologies should be included in revised version.

Round 2

Reviewer 1 Report

Thanks to the careful revision and detailed response made by the authors. All my concerns have been well addressed, and the revised manuscript has been much improved. No other comments. I think this paper deserves to be published in its current form.

Reviewer 2 Report

Authors have addressed all reviewer comments with proper justification and revised manuscript accordingly.